# Hypertension among persons living with HIV—Zambia, 2021; A cross-sectional study of a national electronic health record system

**Jonas Z. Hines**[1]*, **Jose Tomas Prieto**[2], **Megumi Itoh**[1], **Sombo Fwoloshi**[3,4], **Khozya D. Zyambo**[3], **Suilanji Sivile**[3], **Aggrey Mweemba**[3], **Paul Chisemba**[3], **Ernest Kakoma**[3], **Dalila Zachary**[1], **Cecilia Chitambala**[1], **Peter A. Minchella**[1], **Lloyd B. Mulenga**[3], **Simon Agolory**[1]

**1** U.S. Centers for Disease Control and Prevention, Lusaka, Zambia, **2** Palantir Technologies, Paris, France, **3** Ministry of Health, Lusaka, Zambia, **4** University of Zambia, School of Medicine, Lusaka, Zambia

* jhines1@cdc.gov

**Data Availability Statement:** The data that support the findings of this study are available from Zambia Ministry of Health, but restrictions apply to the

## Abstract

Hypertension is a major risk factor for cardiovascular disease, which is a common cause of death in Zambia. Data on hypertension prevalence in Zambia are scarce and limited to specific geographic areas and/or populations. We measured hypertension prevalence among persons living with HIV (PLHIV) in Zambia using a national electronic health record (EHR) system. We did a cross-sectional study of hypertension prevalence among PLHIV aged ≥18 years during 2021. Data were extracted from the SmartCare EHR, which covers ~90% of PLHIV on treatment in Zambia. PLHIV with ≥2 clinical visits in 2021 were included. Hypertension was defined as ≥2 elevated blood pressure readings (systolic ≥140 mmHg/ diastolic ≥90 mmHg) during 2021 and/or on anti-hypertensive medication recorded in their EHR ≤5 years. Logistic regression was used to assess for associations between hypertension and demographic characteristics. Among 750,098 PLHIV aged ≥18 years with ≥2 visits during 2021, 101,363 (13.5%) had ≥2 recorded blood pressure readings. Among these PLHIV, 14.7% (95% confidence interval [CI]: 14.5–14.9) had hypertension. Only 8.9% of PLHIV with hypertension had an anti-hypertensive medication recorded in their EHR. The odds of hypertension were greater in older age groups compared to PLHIV aged 18–29 years (adjusted odds ratio [aOR] for 30–44 years: 2.6 [95% CI: 2.4–2.9]; aOR for 45–49 years: 6.4 [95% CI: 5.8–7.0]; aOR for ≥60 years: 14.5 [95% CI: 13.1–16.1]), urban areas (aOR: 1.9 [95% CI: 1.8–2.1]), and on ART for ≥6-month at a time (aOR: 1.1 [95% CI: 1.0–1.2]). Hypertension was common among PLHIV in Zambia, with few having documentation of treatment. Most PLHIV were excluded from the analysis because of missing BP measurements. Strengthening integrated management of non-communicable diseases in HIV clinics might help to diagnose and treat hypertension in Zambia. Addressing missing data of routine clinical data (like blood pressure) could improve non-communicable diseases surveillance in Zambia.

availability of these data, which were used under license for the current study, and so are not publicly available. Data are however available from the authors upon reasonable request and with permission of Zambia Ministry of Health. To facilitate access, contact the CDC Zambia science office at: zm-ads@cdc.gov.

**Funding:** This research was financially supported by the President's Emergency Plan for AIDS Relief (PEPFAR) through the Centers for Disease Control and Prevention (CDC) through a cooperative agreement with the Zambia Ministry of Health (NU2GGH002234-02) and a grant from the National Institutes of Health to Palantir Technologies (75N95021F00005). Palantir Technologies provided support in the form of salary for JTP. The specific role of this author is articulated in the 'author contributions' section. No additional external funding was received for this study. The funders had no role in study design, data collection and analysis, decision to publish, or preparation of the manuscript.

**Competing interests:** The authors have read the journal's policy and have the following competing interests: JTP is an employee of Palantir Technologies. This does not alter our adherence to PLOS policies on sharing data and materials. There are no patents, products in development or marketed products associated with this research to declare. The authors have declared that no competing interests exist.

## Background

Non-communicable diseases are common causes of death in sub-Saharan African countries [1]. In Zambia, a country in southern Africa, cerebrovascular accidents (CVAs) and ischemic heart disease are among most common causes of death [2,3]. With widespread availability of antiretroviral therapy (ART), non-communicable diseases are increasingly a major cause of morbidity among persons living with HIV (PLHIV) [4,5], including in Zambia [6,7]. Hypertension is a known major risk factor for CVA and other cardiovascular diseases [8]. Although country-specific data on prevalence of hypertension in sub-Saharan African countries is limited, one meta-analysis estimated prevalence to be 16% [9].

PLHIV are at increased risk for noncommunicable diseases including cardiovascular disease [4,10–12]. This risk might be attributed to effects of HIV (i.e., chronic inflammation) and/or side effects (e.g., metabolic syndrome, renal disease) of some ART [13,14]. Some studies indicate PLHIV might have higher prevalence of hypertension than people without HIV [15,16], with roughly 20–25% of PLHIV globally estimated to have hypertension [5,17,18]. However, in sub-Saharan Africa, where most PLHIV reside, hypertension prevalence is not well-characterized at national levels and it is unclear if the prevalence of hypertension differs between PLHIV compared to people without HIV [16,19–21].

Data on hypertension prevalence in Zambia are limited. In the 2018 Zambia Demographic and Health Survey, self-reported hypertension prevalence among women was 8.8% (this information was not ascertained from male participants) [22]. Studies that objectively measure blood pressure have been limited geographically and/or conducted in special populations [7,23–27]. For instance, in one study among PLHIV in Lusaka, hypertension prevalence was 6.4%, of which only one-half of persons were aware of their hypertension diagnosis [24]. Notably, approximately one-quarter of PLHIV with hypertension in that study had suffered a major cardiovascular event, including CVA or myocardial infarction.

Thus, current estimates of hypertension among PLHIV in Zambia give an incomplete picture. In this study, we sought to measure the proportion of PLHIV with hypertension and associate factors using data routinely captured in a national electronic health record (EHR) system, SmartCare EHR.

## Methods

We conducted a cross-sectional study of hypertension among PLHIV aged ≥18 years in Zambia from January to December 2021 (the last full calendar year of data available). We analyzed data from the SmartCare EHR, which was introduced for the HIV program in the early 2000s and has been scaled-up nationally since then. SmartCare EHR supports clinical care by providing patients with their longitudinal health record at any facility operating the EHR. As of 2021, SmartCare EHR was in use in ~1,500 Zambian health facilities that provide care for approximately 90% of PLHIV on ART in Zambia.

Digitized SmartCare EHR data from health facilities are routinely consolidated and de-duplicated at the district and provincial levels, transported to Zambia MOH headquarters in Lusaka, and stored in Zambia's National Data Warehouse. All patient interactions (including clinical, pharmacy, and laboratory visits) at health facilities utilizing SmartCare EHR are recorded and data for most patients are entered into the system in real-time. In cases where health facilities record data on paper forms for retroactive data entry into SmartCare EHR, the process is completed prior to consolidation. Data from inpatient care are not captured in SmartCare EHR.

De-identified demographic data, clinical information, and pharmacy records were extracted from Zambia's SmartCare EHR system. Data were extracted for demographic

characteristics, past medical history, medications, blood pressure measurements, height and weight, and laboratory data (CD4+ count, HIV viral load, and creatinine). Data are cleaned upon ingestion in Foundry (Palantir Technologies, Paris, France) by casting laboratory tests results values and physical measurements to numeric format. For some variables (i.e., non-HIV past medical history, diet, amount of physical activity, smoking history, blood cholesterol levels, and blood glucose) data were either completely missing or too sparse to analyze (i.e., <1% complete). Only PLHIV aged ≥18 years with ≥2 clinical visits in 2021 in SmartCare EHR were eligible for inclusion in the study (in Zambia, BP is measured at each clinical visit but not necessarily at other patient interactions captured in SmartCare EHR [e.g., lab check, pharmacy pick-up]).

We defined elevated blood pressure as a systolic blood pressure reading of ≥140 mmHg or diastolic blood pressure readings of ≥90 mmHg. We defined hypertension as having ≥2 systolic blood pressure (SBP) readings of ≥140 mmHg or ≥2 diastolic blood pressure (DBP) readings of ≥90 mmHg [28] during 2021, or any PLHIV prescribed an antihypertensive medication in SmartCare EHR (including amiloride, amlodipine, atenolol, carvedilol, enalapril, furosemide, hydralazine, hydrochlorothiazide, losartan, methyldopa, metoprolol, nifedipine, spironolactone, telmisartan, and valsartan) in the past five years. Because past medical history was not well captured in the EHR, we could not include persons with a historical hypertension diagnosis regardless of BP measurement readings during 2021. Among PLHIV with hypertension, we defined grade 2 hypertension as ≥1 reading with systolic blood pressure ≥160 mmHg or diastolic blood pressure ≥100 mmHg and hypertensive urgency as ≥1 reading with systolic blood pressure ≥180 mmHg or diastolic blood pressure ≥110 mmHg [28].

Hypertension was measured only among PLHIV with ≥2 blood pressure readings during 2021 because the hypertension case definition required two blood pressure readings. 95% confidence intervals (CIs) were calculated using the Clopper-Pearson exact method in R using epiR package [29]. Bivariable logistic regression was used to measure the association between hypertension and independent variables. We conducted a multivariable logistic regression with variables with ≤10% missingness (i.e., sex, age group, province, urban/rural designation, years on ART, current ART regimen, prescription length, and most recent viral load).

We also conducted an additional age- and sex- adjusted analysis to investigate the relationships between hypertension and kidney function. Specifically, we assessed the relationship between elevated creatinine (i.e., glomerular filtration rate <60 mL/min/1.73m$^2$) in the past year and hypertension. This was a separate analysis because creatinine data were too sparse to include in the multivariable analysis.

Lastly, we also assessed the association between integrase-inhibitor-containing ART regimens and being overweight or having obesity, because of prior association between metabolic syndrome and this medication class. This was relevant to the study objectives given there is an association between metabolic syndrome and cardiovascular disease. Being overweight or having obesity and hypertension were the only two components of metabolic syndrome we could measure from SmartCare EHR; the other components (high triglycerides, low high-density lipoprotein, and elevated fasting glucose) were too sparsely captured. This analysis was age- and sex-adjusted.

## Ethics statement

The study protocol was approved by the ERES Converge IRB in Lusaka, Zambia; it was also reviewed in accordance with CDC human research protection procedures and was determined to be research, but CDC investigators did not interact with human subjects or have access to identifiable data or specimens for research purposes. All methods were carried out in

accordance with relevant guidelines and regulations. This project met requirements for waiver of informed consent documentation, which was granted by ERES Converge IRB in Zambia.

## Results

Among 1,299,263 active PLHIV in SmartCare EHR during 2021, there were 750,098 (57.7%) persons aged ≥18 years that had ≥2 clinical visits in 2021 (Fig 1). Of these, 101,363 (13.5%) had ≥2 blood pressure readings recorded and were included in the analysis.

The complete cohort differed from the analytic cohort for all variables, although absolute differences for key variables like sex and age were minor (e.g., 5.5% were aged ≥60 years in the complete cohort compared to 6.6% in analytic cohort) (Table 1). Health facilities with direct electronic data entry at the point-of-care, which were concentrated in Lusaka and Southern Provinces, had greater blood pressure data completeness than health facilities where data was captured on paper and retrospectively entered into SmartCare EHR (22.5% versus 5.3% captured ≥2 BP readings, respectively).

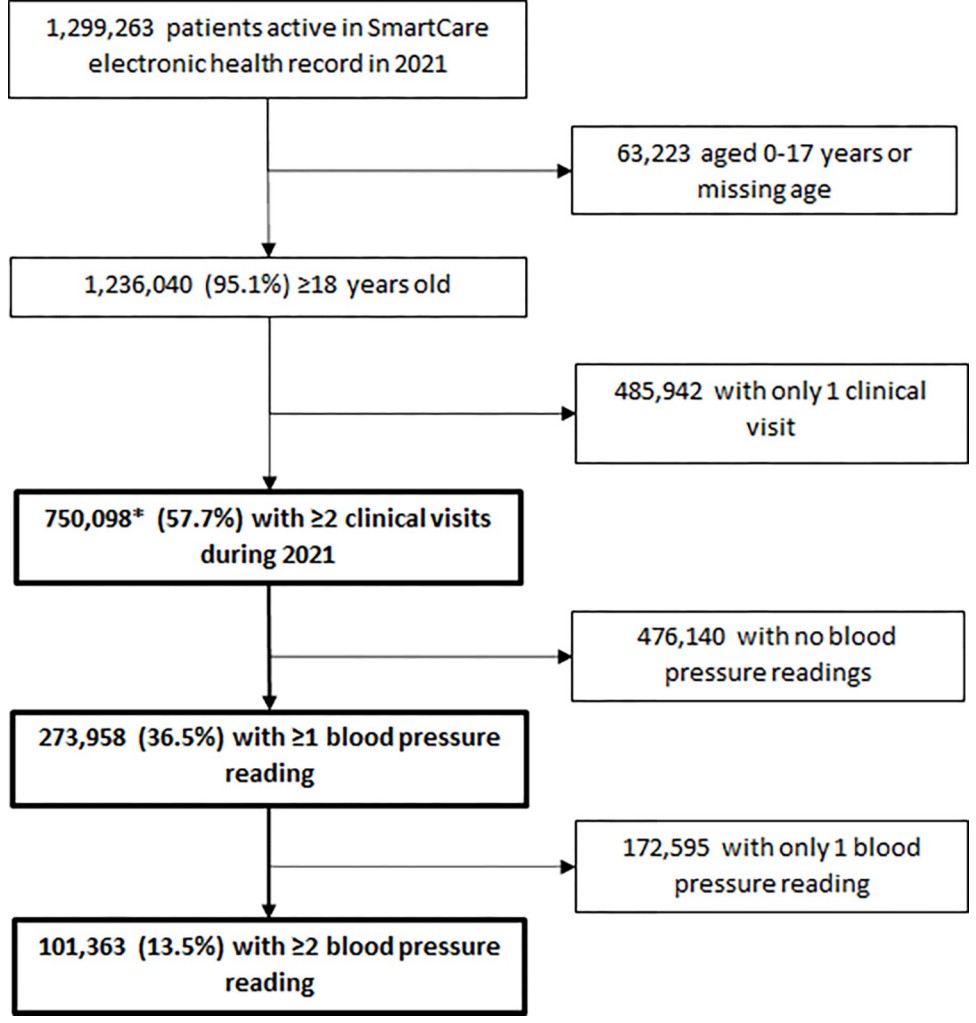

**Fig 1. Sample size flow diagram for analysis of persons living with HIV with hypertension in Zambia, 2021.** * This is the population which was eligible for inclusion in the analysis.

**Table 1. Comparison of the variable distribution and completeness in the entire and analytic datasets for hypertension among persons living with HIV—Zambia, 2021.**

| Variable | Entire dataset, n (%) | Analytic dataset, n (%) | p-value |
|---|---|---|---|
| | (N = 1,236,040) | (N = 101,363) | |
| *Sex* | | | |
| Female | 781,829 (63.3) | 65,570 (64.7) | <0.01 |
| Male | 454,211 (36.7) | 35,793 (35.3) | |
| *Age group* | | | |
| 18–29 years | 232,099 (18.8) | 16,191 (16.0) | <0.01 |
| 30–44 years | 585,455 (47.4) | 45,850 (45.2) | |
| 45–59 years | 350,092 (28.3) | 32,681 (32.2) | |
| ≥60 years | 68,394 (5.5) | 6,641 (6.6) | |
| *Province* | | | |
| Central | 125,704 (10.2) | 2,813 (2.8) | <0.01 |
| Copperbelt | 238,609 (19.3) | 7,197 (7.1) | |
| Eastern | 105,595 (8.5) | 3,672 (3.6) | |
| Luapula | 55,168 (4.5) | 54.0 (0.1) | |
| Lusaka | 343,981 (27.8) | 45,414 (44.8) | |
| Muchinga | 32,978 (2.7) | 512 (0.5) | |
| Northern | 52,014 (4.2) | 58.0 (0.1) | |
| Northwestern | 36,016 (2.9) | 1,893 (1.9) | |
| Southern | 128,299 (10.4) | 35,082 (34.6) | |
| Western | 91,156 (7.4) | 4,120 (4.1) | |
| Missing | 26,520 (2.1) | 548 (0.5) | |
| *Urban/rural designation* | | | |
| Rural | 387,472 (31.3) | 18,454 (18.2) | <0.01 |
| Urban | 736,921 (59.6) | 78,123 (77.1) | |
| Missing | 111,647 (9.0) | 4,786 (4.7) | |
| *Years on ART* | | | |
| 0–1 | 340,014 (27.5) | 19,087 (18.8) | <0.01 |
| 2–4 | 419,400 (33.9) | 28,182 (27.8) | |
| 5–9 | 301,575 (24.4) | 32,408 (32.0) | |
| ≥10 | 175,051 (14.2) | 21,686 (21.4) | |
| *Current ART regimen* | | | |
| Efavirenz-based | 49,083 (4.0) | 1,423 (1.4) | <0.01 |
| Dolutegravir-based | 1,137,837 (92.1) | 97,049 (95.7) | |
| Both efavirenz and dolutegravir listed | 3,291 (0.3) | 133 (0.1) | |
| Other | 45,829 (3.7) | 2,758 (2.7) | |
| *Most recent ART prescription length* | | | |
| <3 months | 141,776 (11.5) | 6,275 (6.2) | <0.01 |
| 3–5 months | 373,549 (30.2) | 27,839 (27.5) | |
| 6+ months | 720,706 (58.3) | 67,249 (66.3) | |
| Missing | 9.00 (0.0) | 0 (0.0) | |
| *Body mass index (kg/m$^2$)* | | | |
| Normal (18.5–24.9) | 336,236 (27.2) | 51,056 (50.4) | <0.01 |
| Low (<18.5) | 59,860 (4.8) | 8,776 (8.7) | |
| Overweight (25.0–29.9) | 107,253 (8.7) | 19,668 (19.4) | |
| Obesity (≥30.0) | 53,674 (4.3) | 10,585 (10.4) | |
| Missing | 679,017 (54.9) | 11,278 (11.1) | |

*(Continued)*

**Table 1.** (Continued)

| Variable | Entire dataset, n (%) | Analytic dataset, n (%) | p-value |
|---|---|---|---|
| | **(N = 1,236,040)** | **(N = 101,363)** | |
| *Initial CD4+ count (cells/mm$^3$)* | | | |
| 0–200 | 62,431 (5.1) | 7,694 (7.6) | <0.01 |
| 201–350 | 61,544 (5.0) | 7,195 (7.1) | |
| >350 | 107,473 (8.7) | 12,847 (12.7) | |
| Missing | 100,4592 (81.3) | 73,627 (72.6) | |
| *Most recent CD4+ count (cells/mm$^3$)** | | | |
| 0–200 | 59,590 (4.8) | 6,994 (6.9) | <0.01 |
| 201–350 | 101,297 (8.2) | 12,567 (12.4) | |
| >350 | 369,859 (29.9) | 49,690 (49.0) | |
| Missing | 705,294 (57.1) | 32,112 (31.7) | |
| *Most recent viral load (copies/mL)* † | | | |
| <1000 | 975,952 (79.0) | 92,115 (90.9) | <0.01 |
| 1,000–9,999 | 16,881 (1.4) | 1,494 (1.5) | |
| ≥10,000 | 26,338 (2.1) | 2,595 (2.6) | |
| Missing | 216,869 (17.5) | 5,159 (5.1) | |
| *Most recent creatinine‡* | | | |
| Normal | 36,457 (2.9) | 8,892 (8.8) | <0.01 |
| High | 5,829 (0.5) | 1,254 (1.2) | |
| Missing | 119,3754 (96.6) | 91,217 (90.0) | |
| *Data capture at point-of-care* | | | |
| Paper-based with retrospective input | 495,498 (40.1) | 73,406 (72.4) | <0.01 |
| Direct electronic input | 552,109 (44.7) | 17,197 (17.0) | |
| Missing | 188,433 (15.2) | 10,760 (10.6) | |

Variables are based on data from the most recent patient interaction in the electronic health record.

* 38.9% of recent CD4+ count measurements were from 2020 or 2021.

† 98.4% of recent viral load measurements were from 2020 or 2021.

‡ Elevated creatinine defined as ≥115 µmol/L in men and ≥98 µmol/L in women.

ART: Antiretroviral therapy.

Among PLHIV in the analytic dataset, the mean age was 41.6 years (standard deviation ±11.4 years; range 18–104 years) and 64.7% were females. The median time on ART was 5.0 years (interquartile range: 2.0–9.0 years) and 95.7% of PLHIV were on dolutegravir-based ART regimens at most recent visit.

During 2021, 35.0% of PLHIV had ≥1 elevated blood pressure reading. During 2021, 14.7% (95% CI: 14.5–14.9) of PLHIV had hypertension (Table 2). The proportion of PLHIV with hypertension increased with increasing age, from 4.3% among PLHIV aged 18–29 years, to 10.1% among PLHIV aged 30–44 years, to 21.6% among PLHIV aged 45–59 years, and 37.4% for among PLHIV aged ≥60 years. Among PLHIV with hypertension, 60.7% had grade 2 hypertension and 27.0% had hypertensive urgency.

Overall, 2.0% of PLHIV had an anti-hypertensive medication recorded in their EHR in the past five years; 8.9% of PLHIV with two or more readings of SBP ≥140 mmHg or DBP ≥90 mmHg had an anti-hypertensive medication recorded in their EHR. Among any PLHIV with an anti-hypertensive medication recorded, 85.9% had ≥1 reading with SBP ≥140 mmHg or DBP ≥90 mmHg and 60.2% had two elevated readings (i.e., were still hypertensive).

**Table 2.  Hypertension prevalence and odds ratios of hypertension among persons living with HIV—Zambia, 2021 (N = 101,363)\*.**

| | Prevalence, % | OR (95% CI) | aOR (95% CI)† |
|---|---|---|---|
| Overall | 14.7 | | |
| Sex | | | |
| Female | 13.7 | Referent | Referent |
| Male | 16.4 | 1.23 (1.19–1.28) | 1 (0.96–1.04) |
| Age group | | | |
| 18–29 years | 4.3 | Referent | Referent |
| 30–44 years | 10.1 | 2.53 (2.33–2.75) | 2.61 (2.39–2.85) |
| 45–59 years | 21.6 | 6.19 (5.71–6.71) | 6.36 (5.81–6.96) |
| ≥60 years | 37.4 | 13.42 (12.26–14.7) | 14.54 (13.14–16.09) |
| Province | | | |
| Central | 11.8 | Referent | Referent |
| Copperbelt | 18.1 | 1.65 (1.45–1.88) | 1.25 (1.08–1.44) |
| Eastern | 10.3 | 0.87 (0.74–1.01) | 0.62 (0.5–0.78) |
| Luapula | 5.6 | 0.44 (0.14–1.42) | 0.4 (0.09–1.67) |
| Lusaka | 15.4 | 1.37 (1.22–1.54) | 1.11 (0.97–1.26) |
| Muchinga | 3.7 | 0.29 (0.18–0.46) | 0.37 (0.21–0.63) |
| Northern | 10.3 | 0.87 (0.37–2.03) | 1.23 (0.42–3.58) |
| Northwestern | 12.4 | 1.06 (0.88–1.26) | 0.92 (0.76–1.11) |
| Southern | 14.6 | 1.28 (1.14–1.44) | 0.97 (0.85–1.1) |
| Western | 8.9 | 0.73 (0.62–0.85) | 0.78 (0.66–0.92) |
| Urban/rural designation | | | |
| Rural | 9.0 | Referent | Referent |
| Urban | 16.0 | 1.92 (1.82–2.03) | 1.93 (1.81–2.05) |
| Years on ART | | | |
| 0–1 | 10.8 | Referent | Referent |
| 2–4 | 12.8 | 1.21 (1.14–1.28) | 1.04 (0.97–1.11) |
| 5–9 | 15.0 | 1.46 (1.38–1.54) | 1.02 (0.95–1.08) |
| ≥10 | 20.1 | 2.09 (1.97–2.21) | 1.07 (1.00–1.14) |
| ART regimen‡ | | | |
| Efavirenz-based | 11.5 | Referent | Referent |
| Dolutegravir-based | 14.8 | 1.33 (1.13–1.57) | 1.15 (0.96–1.38) |
| Other | 13.7 | 1.22 (1.00–1.48) | 1.00 (0.80–1.24) |
| Most recent ART prescription length | | | |
| <3 months | 11.3 | Referent | Referent |
| 3–5 months | 12.6 | 1.14 (1.04–1.24) | 0.98 (0.89–1.09) |
| ≥6 months | 15.9 | 1.48 (1.36–1.60) | 1.11 (1.01–1.22) |
| Body mass index (kg/m²) | | | |
| Normal (18.5–24.9) | 11.6 | Referent | |
| Low (<18.5) | 8.6 | 0.72 (0.66–0.78) | |
| Overweight (25–29.9) | 19.9 | 1.9 (1.82–1.99) | |
| Obesity (≥30.0) | 28.7 | 3.08 (2.93–3.24) | |
| Initial CD4+ count (cells/mm³) | | | |
| 0–200 | 15.2 | Referent | |
| 201–350 | 15.0 | 0.98 (0.9–1.07) | |
| >350 | 13.6 | 0.87 (0.81–0.95) | |
| Most recent CD4+ count (cells/mm³) | | | |
| 0–200 | 15.2 | Referent | |

(Continued)

**Table 2.** (Continued)

|  | Prevalence, % | OR (95% CI) | aOR (95% CI)[†] |
|---|---|---|---|
| 201–350 | 17.0 | 1.15 (1.06–1.24) |  |
| >350 | 16.2 | 1.09 (1.01–1.16) |  |
| *Most recent viral load count (copies/mL)* |  |  |  |
| <1000 | 15.1 | Referent | Referent |
| 1,000–9,999 | 11.5 | 0.73 (0.62–0.86) | 0.96 (0.81–1.14) |
| ≥10,000 | 10.9 | 0.69 (0.61–0.78) | 1.02 (0.89–1.17) |
| *Most recent creatinine* [¶] |  |  |  |
| Normal creatinine | 17.3 | Referent |  |
| Elevated creatinine | 28.1 | 1.88 (1.64–2.15) |  |

* Hypertension defined as ≥2 systolic blood pressure readings of ≥140 mmHg or ≥2 diastolic blood pressure readings of ≥90 mmHg among persons with ≥2 clinical visits during 2021.

† Adjusted for sex, age group, province, urban/rural designation, ART regimen, years on ART, script length, and body mass index.

‡ The most recently listed ART regimen in SmartCare electronic health record. If a regimen listed both dolutegravir and efavirenz, then it was excluded from the analysis (n = 128).

¶ Elevated creatinine defined as ≥115 μmol/L in men and ≥98 μmol/L in women, corresponding to glomerular filtration rate of <60 mL/min/1.73m$^2$.

aOR: Adjusted odds ratio; ART: Antiretroviral therapy; CI: Confidence interval; OR: Odds ratio.

In the multivariable model, the odds of hypertension were greater for older age groups, PLHIV residing in urban areas and certain provinces, and PLHIV prescribed ART for ≥6-month at a time (Table 2). Although dolutegravir-based regimens were associated with higher odds of hypertension compared to efavirenz-based regimens in the bivariable analysis, there was no difference after adjustment in the multivariable model. PLHIV who were overweight or had obesity had greater odds of hypertension than normal weight PLHIV (although data missingness precluded inclusion of this characteristic in the multivariable model).

PLHIV with hypertension had greater odds of having an elevated creatinine (Table 3). Lastly, PLHIV on dolutegravir-based regimens had higher odds of being overweight or having obesity compared to persons on other regimens (adjusted OR: 1.16 [95% CI: 1.03–1.32]).

**Table 3. Prevalence and odds ratios of elevated creatine among persons living with HIV—Zambia, 2021 (N = 10,146)\*.**

|  | Prevalence, % | OR (95% CI) | aOR (95% CI)[†] |
|---|---|---|---|
| Overall | 12.4 |  |  |
| Blood pressure |  |  |  |
| Not hypertensive[‡] | 10.9 | Referent | Referent |
| Hypertensive[‡] | 18.7 | 1.88 (1.64–2.15) | 1.36 (1.18–1.57) |
| Sex |  |  |  |
| Female | 11.1 | Referent | Referent |
| Male | 14.4 | 1.35 (1.2–1.52) | 1.19 (1.06–1.35) |
| Age Group |  |  |  |
| 18–29 years | 4.8 | Referent | Referent |
| 30–44 years | 10.2 | 2.28 (1.8–2.89) | 2.21 (1.75–2.81) |
| 45–59 years | 15.7 | 3.72 (2.95–4.69) | 3.38 (2.67–4.28) |
| ≥60 years | 27.0 | 7.42 (5.69–9.68) | 6.37 (4.85–8.37) |

* Elevated creatinine defined as ≥115 μmol/L in men and ≥98 μmol/L in women, corresponding to glomerular filtration rate of <60 mL/min/1.73m$^2$.

† Adjusted for presence/absence of hypertension, sex, and age group.

‡ Hypertension defined as as having ≥2 systolic blood pressure readings of ≥140 mmHg or ≥2 diastolic blood pressure readings of ≥90 mmHg during the study period.

aOR: Adjusted odds ratio; CI: Confidence interval; OR: Odds ratio.

## Discussion

Hypertension was common among PLHIV in Zambia, with one in about every seven PLHIV aged ≥18 years having hypertension during a one-year period. This estimate is similar to other studies of PLHIV in sub-Saharan Africa, if not slightly lower [12,30]. Most PLHIV with hypertension had dangerously high blood pressure readings (i.e., grade 2 hypertension) putting them at elevated risk for cardiovascular disease including CVAs and acute cardiovascular events. This finding might explain why cardiovascular disease are among the most common causes of death among PLHIV in Zambia [6,31,32]. Few PLHIV with hypertension had documentation of being on antihypertensive treatment and among those that were, most did not have their blood pressure under control (as demonstrated by elevated blood pressure readings in these patients). Other studies in Zambia indicate suboptimal levels of hypertension treatment and control [22,24], which is similar to other countries in Africa [21,33,34]. Integrating noncommunicable disease care into routine HIV care might increase prevention, diagnosis, and management of hypertension in Zambia, potentially reducing cardiovascular disease-related morbidity and mortality [35]. Although most PLHIV were excluded from the study because of missing BP data, to our knowledge, this analysis is the largest cohort study of hypertension among PLHIV.

That many hypertensive PLHIV still had elevated blood pressure readings despite antihypertensive treatment is demonstrative of the challenge of controlling hypertension even when treated [21,33,34]; nevertheless this finding warrants action, with a focus on strategies to increase treatment of PLHIV with existing hypertension in Zambia and measures to prevent hypertension among those without it. Being older and overweight are established risk factors for hypertension, including among PLHIV [12,21]. The observed geographic patterns of hypertension could be related to urban/rural differences in environmental factors associated with hypertension (i.e., consumption of unhealthy diets such as diets high in sodium, exposure to pollution, or reduced opportunities for physical activity) and/or be an artifact of poor data quality (i.e., highest proportion of PLHIV with hypertension observed in provinces that also had better data completeness). The factors associated with hypertension in this analysis mirror those among the general population in countries in Africa [36].

Some reports link ART use with increased prevalence of hypertension among PLHIV in Africa [7,37,38]. Although we could not assess this relationship in this study that was confined to PLHIV on ART, longer ART duration was not associated with hypertension, which is in contrast to findings from several other studies from countries in Africa [18,39]. This could reflect the impact of the 'test-and-treat' strategy that was introduced in Zambia ~2016, resulting in earlier viral suppression and thus reduced consequences of uncontrolled chronic HIV infection. Some types of ART have been associated with hypertension [7,40], but in this analysis there were no associations between ART regimens and hypertension in the multivariable model. However, those on longer prescription duration did have higher hypertension prevalence, which could be a result of less frequent contact with the health care system and therefore the opportunity for chronic conditions like hypertension to go undiagnosed and untreated or, alternatively, residual age confounding (i.e., older patients are more likely to be on stable HIV treatment [41].

The association of elevated BMI for PLHIV on dolutegravir-based regimens in this analysis could signify metabolic syndrome among these persons, with a potential side effect of integrase inhibitors [42]; we were not able to analyze glucose or lipid measurements to confirm this hypothesis. However, because many PLHIV in Zambia were transitioned to dolutegravir-based regimens from different ART regimens (i.e., efavirenz-based regimens) in the recent past, the elevated BMI among participants could also pre-date their transition to dolutegravir-

based regimens so a different study design (i.e., cohort study) is warranted to further explore this potential signal. Nevertheless, the superior HIV viral load control and lower risk of HIV treatment failure make dolutegravir the preferred regimen in Zambia and other countries with generalized HIV epidemics [43].

The study had several limitations. Most importantly, blood pressure data completeness was very low, with only approximately one-eighth of PLHIV in the dataset being analyzed. Despite this limitation, this is one of the largest studies of hypertension reported from sub-Saharan Africa to date and provides the first national level study of hypertension in Zambia. However, the estimate is not nationally representative and, furthermore, only represents an estimate among PLHIV in care who were captured by the SmartCare EHR. Furthermore, the data for this analysis are mostly from urban facilities in Lusaka and Southern Provinces. Comparison to non-HIV-infected persons was not possible, but as SmartCare EHR is integrated into other care settings in Zambia, this will become possible. For some variables, high amounts of missingness precluded their inclusion in the multivariable model and, furthermore, some important variables (e.g., non-HIV past medical history) were not available. Additionally, past medical history was not captured in the dataset, so the proportion of PLHIV with diagnosed hypertension could not be assessed. Furthermore, very few records had a blood pressure medication documented. This could reflect low levels of hypertension treatment, limited availability of sphygmomanometers, or could result from data entry omissions at the point-of-care. If antihypertensive medications were not consistently recorded, then the true prevalence of hypertension among PLHIV in Zambia is likely higher. Next, only blood pressure readings occurring over a one-year period were assessed. This approach reduced the likelihood that two elevated blood pressure measurements were separated by long periods of time, but also led to the exclusion of blood pressure measurements that occurred outside of the defined period potentially affecting the hypertension estimates in the analysis. Lastly, the EHR does not capture inpatient data, so information on consequences of uncontrolled hypertension (e.g., stroke or myocardial infarction) were not available.

This analysis points to a need to improve hypertension management for PLHIV in Zambia. Some of the existing practices that rely on referring patients with elevated BPs to the outpatient department (i.e., urgent care) for further evaluation might result in patient attrition, missing opportunities to adequately manage this common comorbidity among PLHIV in Zambia. Integrated primary care models for PLHIV have better outcomes for non-communicable disease management [44], and can even result in better viral suppression [45]. ART clinics in Zambia might benefit from instituting integrated management of noncommunicable diseases, including life-style modification, anti-hypertensive therapy with appropriate treatment intensification, and medication adherence assessments for hypertension [46]. An integrated primary care model for conditions like hypertension is possible in countries like Zambia and can improve patient outcomes [34,35]. With integration, multi-month dispensing is a promising approach for co-management of HIV and hypertension [47].

Routinely monitoring for hypertension, along with documenting other health measures (e.g., diet, smoking status, BMI) in the EHR would not only allow medical providers to determine client's cardiovascular risk for CVAs or other cardiovascular events and target treatment, but also might provide data needed to identify specific health facilities or clinicians that would benefit from educational interventions in management of these risk factors and conditions. EHRs like SmartCare are promising data sources for noncommunicable diseases surveillance given their reach, routine use in clinical setting, and richness of information. This analysis suggests EHRs are being more widely implemented and adopted in urban areas, which has the potential to affect urban-rural differences in surveillance and/or care. For this to be successful, data completeness needs to be improved to routinely capture cardiovascular disease risk factors, including blood pressure readings consistently for PHLIV in their EHRs.

Hypertension was common among PLHIV in Zambia and many persons might not be adequately diagnosed or treated. It is important for ART clinic providers to consider hypertension among PLHIV and institute strategies to manage it appropriately. This will require adequate capacitation of the Zambian health workforce to recognize and manage hypertension. Additionally, care models that integrate hypertension (and other NCDs) management into ART clinics are promising strategies to improve care. EHRs might be used to routinely track program implementation at little additional data collection effort, and can be adapted into noncommunicable diseases surveillance systems. Addressing hypertension and other noncommunicable diseases will be important to reducing morbidity and mortality of PLHIV in Zambia.

## Supporting information

**S1 File. Scientific poster entitled, "Hypertension prevalence among persons living with HIV—Zambia, July 2020–June 2021," presented at the conference on retroviruses and opportunistic infection in February 2022.** Available from: https://www.croiconference.org/abstract/hypertension-prevalence-among-persons-living-with-hiv-zambia-july-2020-june-2021/.
(PDF)

**S1 Text. PLOS inclusivity in global research questionnaire.**
(DOCX)

## Acknowledgments

**Authorship Disclaimer:** The findings and conclusions in this report are those of the author(s) and do not necessarily represent the official position of the funding agencies.

## Author Contributions

**Conceptualization:** Jonas Z. Hines.

**Data curation:** Jonas Z. Hines, Jose Tomas Prieto, Cecilia Chitambala, Peter A. Minchella.

**Formal analysis:** Jonas Z. Hines.

**Methodology:** Jonas Z. Hines, Jose Tomas Prieto, Peter A. Minchella.

**Supervision:** Simon Agolory.

**Writing – original draft:** Jonas Z. Hines.

**Writing – review & editing:** Jose Tomas Prieto, Megumi Itoh, Sombo Fwoloshi, Khozya D. Zyambo, Suilanji Sivile, Aggrey Mweemba, Paul Chisemba, Ernest Kakoma, Dalila Zachary, Cecilia Chitambala, Peter A. Minchella, Lloyd B. Mulenga, Simon Agolory.

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
