## [Decision Letter · Decision Letter 0]

20 Mar 2023

PGPH-D-23-00227

Hypertension among persons living with HIV — Zambia, 2021; A cross-sectional study of a national electronic health record system

Dear Dr. Hines,

Thank you for submitting your manuscript to PLOS Global Public Health. After careful consideration, we feel that it has merit but does not fully meet PLOS Global Public Health’s publication criteria as it currently stands. Therefore, we invite you to submit a revised version of the manuscript that addresses the points raised during the review process.

You have observed that your findings in the multivariable model suggest the increased odds of hypertension among older age groups in PLHIV. This intuitive finding is similar to what is commonly observed in the general population. While this is an important observation, your study could benefit from further analysis, utilizing the advantage of a large sample size. For example, are there any prognostic variables associated with higher hypertension apart from age, specifically in PLHIV? Additional analyses to identify and present specific attributes related to hypertension in PLHIV could provide additional insights into the complex relationship between HIV and hypertension. Also, please expand the discussion section of your manuscript to include a comparison of your study findings with the general population (non-PLHIV). This comparison should highlight the similarities and contrasts between findings in PLHIV and the general population and provide a broader context for understanding the implications of the results.

We look forward to receiving your revised manuscript.

Kind regards,

Giridhara R Babu, MBBS, MPH, PhD

Academic Editor

Journal Requirements:

2. Please include the following request in the decision letter, and ping me with follow up. “Please include a complete copy of PLOS’ questionnaire on inclusivity in global research in your revised manuscript. Our policy for research in this area aims to improve transparency in the reporting of research performed outside of researchers’ own country or community. The policy applies to researchers who have travelled to a different country to conduct research, research with Indigenous populations or their lands, and research on cultural artefacts. The questionnaire can also be requested at the journal’s discretion for any other submissions, even if these conditions are not met.  Please find more information on the policy and a link to download a blank copy of the questionnaire here: https://journals.plos.org/plosone/s/best-practices-in-research-reporting. Please upload a completed version of your questionnaire as Supporting Information when you resubmit your manuscript.”

3. Please amend your detailed Financial Disclosure statement. This is published with the article. It must therefore be completed in full sentences and contain the exact wording you wish to be published.

4. Please provide separate figure files in .tif or .eps format only and remove any figures embedded in your manuscript file. Please also ensure that all files are under our size limit of 10MB.

5. We have noticed that you have uploaded Supporting Information files, but you have not included a list of legends. Please add a full list of legends for your Supporting Information files after the references list.

Additional Editor Comments (if provided):

Reviewers' comments:

Reviewer's Responses to Questions

**Comments to the Author**

1. Does this manuscript meet PLOS Global Public Health’s publication criteria? Is the manuscript technically sound, and do the data support the conclusions? The manuscript must describe methodologically and ethically rigorous research with conclusions that are appropriately drawn based on the data presented.

Reviewer #1: Partly

Reviewer #2: Partly

Reviewer #3: Yes

2. Has the statistical analysis been performed appropriately and rigorously?

Reviewer #1: Yes

Reviewer #2: Yes

Reviewer #3: Yes

3. Have the authors made all data underlying the findings in their manuscript fully available (please refer to the Data Availability Statement at the start of the manuscript PDF file)?

Reviewer #1: No

Reviewer #2: Yes

Reviewer #3: Yes

4. Is the manuscript presented in an intelligible fashion and written in standard English?

Reviewer #1: Yes

Reviewer #2: Yes

Reviewer #3: Yes

5. Review Comments to the Author

Reviewer #1: This is an insightful study that will enable strengthening of health service delivery to PLHIV and other health system components such as Health Information System: Data entry, storage and capacity building among health care workers.

A number of issues to consider:

1) The use of the term 'common' in the first sentence in the discussion both in the abstract and discussion section (main body) could suggest high prevalence. The study findings show a prevalence of about 15%, compared to the prevalence rates of hypertension in Africa (16%), the global rates among PLHIV (20-25%) and Zambian DHS (8.8%) as presented in the introduction section. A comparator could complement the statement and increase clarity. Also, do we have statistics on hypertension among PLHIV in Africa or Southern Africa region, inclusion of the same, could help show the magnitude of the hypertension among the population besides the general population statistics.

2) STUDY OBJECTIVE: The objective stated in line 83-85 should collate to the methodology and other parts of the results. As presented in lines 134-145, the study also aimed to assess associated factors to the hypertension in PLHIV besides measuring the proportion of hypertension. As stated in the background section it only indicates the study as a prevalence study.

3) Clarification on hypertension case definition in line 127. The hypertension diagnosis requires 'at least' 2 readings.

4) Use of the term 'many' in line 189

5) Line 208 'high sodium diet'. Consider rephrasing e.g., consumption of diets high in sodium or consumption of unhealth diets such as diets high in sodium to improve clarity on the plausible exposing factors, since there are other dietary factors that also predispose the population to hypertension or cardiovascular risk factors.

6) Study limitation; consider adding a limitation on how most of the data is from urban facilities (77%) and from Lusaka and Southern provinces (79%) limits the interpretation of the results.

7) The article could also highlight the implication of data availability. As presented in table 1, most the data is from urban facilities, and how it could be important to strengthen the health information system to support health service delivery and practice (health care).

8) Capitalise 'all' in line 285

Reviewer #2: Thank you for this scientific work which reflects the data gaps in electronic surveillance systems in developing countries. However, please note in the Method: the TYPE OF CROSS-SECTIONAL STUDY (DESCRIPTIVE OR ANALYTICAL) you have carried out; the type of sample and the sampling technique used. In addition, data such as the mean, median and measure of variance are not available in the tables.

Reviewer #3: Very well conducted. Comprehensive research study.

Only addition can be if researchers can highlight on any peculiarity in PLHIV cohort as compared to general population in trend of hypertension otherwise age bracket, gender, urban-rural divide are giving the same trend as of general population.

6. PLOS authors have the option to publish the peer review history of their article (what does this mean?). If published, this will include your full peer review and any attached files.

**Do you want your identity to be public for this peer review?** For information about this choice, including consent withdrawal, please see our Privacy Policy.

Reviewer #1: No

Reviewer #2: **Yes: **Carlos TIEMENI

Reviewer #3: **Yes: **Dr. Shrikant Kishorrao Kalaskar

---

## [Decision Letter · Decision Letter 1]

14 Jun 2023

Hypertension among persons living with HIV — Zambia, 2021; A cross-sectional study of a national electronic health record system

PGPH-D-23-00227R1

Dear Dr. Hines,

We are pleased to inform you that your manuscript 'Hypertension among persons living with HIV — Zambia, 2021; A cross-sectional study of a national electronic health record system' has been provisionally accepted for publication in PLOS Global Public Health.

Best regards,

Giridhara R Babu, MBBS, MPH, PhD

Academic Editor

Reviewer Comments (if any, and for reference):

Reviewer's Responses to Questions

**Comments to the Author**

1. If the authors have adequately addressed your comments raised in a previous round of review and you feel that this manuscript is now acceptable for publication, you may indicate that here to bypass the “Comments to the Author” section, enter your conflict of interest statement in the “Confidential to Editor” section, and submit your "Accept" recommendation.

Reviewer #1: All comments have been addressed

Reviewer #2: All comments have been addressed

Reviewer #3: All comments have been addressed

2. Does this manuscript meet PLOS Global Public Health’s publication criteria? Is the manuscript technically sound, and do the data support the conclusions? The manuscript must describe methodologically and ethically rigorous research with conclusions that are appropriately drawn based on the data presented.

Reviewer #1: Yes

Reviewer #2: Yes

Reviewer #3: Yes

3. Has the statistical analysis been performed appropriately and rigorously?

Reviewer #1: Yes

Reviewer #2: Yes

Reviewer #3: Yes

4. Have the authors made all data underlying the findings in their manuscript fully available (please refer to the Data Availability Statement at the start of the manuscript PDF file)?

Reviewer #1: Yes

Reviewer #2: Yes

Reviewer #3: Yes

5. Is the manuscript presented in an intelligible fashion and written in standard English?

Reviewer #1: Yes

Reviewer #2: Yes

Reviewer #3: Yes

6. Review Comments to the Author

Reviewer #1: (No Response)

Reviewer #2: (No Response)

Reviewer #3: Thanks for addressing most of the comments. However, if the study is concluding there are more similarities than differences between PLHIV and general population, it is more required to highlight the differences like if average age of diagnosis is more or less given the frequent visits of PLHIV to ART clinics or is there a better control in PLHIV compared to general population?

Adherence to medication compared to general population. Odds of complication as compared to general population?

If the study is not commenting on peculiarities and differences in PLHIV and general population, then there is no difference between any other hypertension prevalence study on general population and this study.

7. PLOS authors have the option to publish the peer review history of their article (what does this mean?). If published, this will include your full peer review and any attached files.

**Do you want your identity to be public for this peer review?** For information about this choice, including consent withdrawal, please see our Privacy Policy.

Reviewer #1: **Yes: **Moriasi Nyanchoka

Reviewer #2: **Yes: **Carlos TIEMENI

Reviewer #3: **Yes: **Dr. Shrikant K Kalaskar
